# Synthesis, Characterization and Anticancer Efficacy Evaluation of Benzoxanthone Compounds toward Gastric Cancer SGC-7901

**DOI:** 10.3390/molecules27061970

**Published:** 2022-03-18

**Authors:** Yuan Fu, Yunran Xu, Yunjun Liu, Yi Wang, Ju Chen, Xiuzhen Wang

**Affiliations:** School of Pharmacy, Guangdong Pharmaceutical University, Guangzhou 510006, China; ssssfu@126.com (Y.F.); xuyunran0210@163.com (Y.X.); wangyi19992021@163.com (Y.W.); chenju152@163.com (J.C.)

**Keywords:** benzoxanthones, gastric cancer cells, apoptosis, cytotoxicity, photochemical synthesis

## Abstract

Three benzoxanthone derivatives were synthesized through a new photochemical strategy. The in vitro cytotoxic activity of these compounds was evaluated by 3-(4,5-dimethylthiazol-2-yl)-2,5-diphenyltetrazolium bromide (MTT) assay, and their partition coefficients (log*P*) were measured by shake flask method. The p*K_a_* values of the compounds were detected by potentionmetric titration. The interaction of the compounds with calf thymus DNA (CT-DNA) was investigated by electronic absorption, luminescence spectra and viscosity. A molecular docking analysis was performed. The antitumor efficacy of the compounds was evaluated by cell apoptosis, cell cycle arrest, intracellular Ca^2+^ concentrations and reactive oxygen species (ROS) levels. The mitochondrial membrane potential was assayed using JC-1 (5,5′,6,6′-tetrachloro-1,1,3′,3′-tetraethyl-imidacarbocyanine iodide) as the fluorescence probe. The expression of Bcl-2 family protein, caspase 3 and poly ADP-ribose polymerase (PARP) was explored by western blot. The results showed that the compounds induced apoptosis through a ROS-mediated mitochondrial dysfunction pathway. This work provides an efficient approach to synthesize benzoxanthone derivatives, and is helpful for understanding the apoptotic mechanism.

## 1. Introduction

Xanthones are the second metabolites from higher plants and microorganisms [1]. Benzoxanthone is a xanthone derivative which possesses antimicrobial, antiviral, antitumor activities [2,3,4]. Because of the diverse pharmacological activities of xanthones, these compounds have received a great deal of attention from researchers. Up to now, many benzoxanthone analogues for prospective drug candidates have been obtained [5,6,7,8]. Some benzoxanthone derivatives have been used as topoisomerase inhibitors and DNA cross-linkers [9,10]. Hurley synthesized bisfuranoxanthones and found the compounds possessed high antitumor activity [11]. You et al. designed a number of hydroxyxanthone derivatives and found an orally active antitumor agent [12]. In general, the synthetic methods for benzoxanthone include the displacement of bromides or iodides through nucleophilic aromatic substitution [13], demethylation with boron tribromide [14], photooxidative cyclization of 2-styryl-chromones [15,16] and the ultraviolet radiation of benzylidene-1-tetralone [17]. Synthetic methods for the production of the compounds in Refs. [13,14,15,16,17] can be seen in the Figure 1. However, these methods have some disadvantages, such as harsh reaction conditions, long reaction times and the need for expensive catalysts. To overcome these drawbacks, we developed a new photochemical method to synthesize dibenzoxanthenes, namely, dibenzoxanthenes underwent ring-opening reactions, proton migration and nucleophilic addition to form carboxy propyl substituted benzoxanthone under illumination conditions. Because no catalyst, no heat and no other solvents were used, we consider that this is a green and effective synthetic method.

Study on the interaction between small molecules and DNA is important to understand the antitumor mechanisms of the former [18]. In this work, we investigated the interaction of benzoxanthone with calf thymus DNA (CT-DNA) by electronic absorption titration, luminescence spectra and a viscosity experiment. It is well known that apoptosis includes two major pathways, i.e., extrinsic (death receptor) and intrinsic (mitochondrial). While the mitochondrial pathway is involved in Bid cleavage and change of the ratio of Bax/Bcl-2, they cause loss of mitochondrial membrane potential and the release of cytochrome c; additionally, caspase 3 and caspase 9 are activated, resulting in apoptosis [19,20,21]. In our previous work, we discovered that dibenzoxanthenes induced apoptosis through the ROS-mediated mitochondrial dysfunction pathway [22]. To obtain more information on the anticancer activity of dibenzoxanthenes, in this paper, three dibenzoxanthene derivatives (Figure 2) were synthesized and characterized by HRMS, ^1^H NMR and ^13^C NMR. Their anticancer activity was explored by cytotoxicity, apoptosis, intracellular ROS levels and the change of mitochondrial membrane potential. Additionally, we also investigated the interactions of the dibenzoxanthene derivatives with calf thymus DNA (CT-DNA).

## 2. Materials and Methods

### 2.1. Materials

A Bruker AVANCE-500 spectrometer was used to detect NMR spectra. All chemical shifts were given relative to tetramethylsilane (TMS). Bruker 7.0 T SolariX XR FT-ICR-MS was used to record the Mass spectra. TLC-analysis was performed on glass-backed plates (Merck) coated with 0.2 mm silica 60F254. Commercial common reagent-grade chemicals were used without further purification. The gastric adenocarcinoma cell line SGC-7901, the cervical cancer cell line HeLa, lung carcinoma cell line A549, the human hepatocellular carcinoma cell line BEL-7402, the normal cell line LO2 were purchased from the cell bank of the Cell Institute of Sinica Academia Shanghai (Shanghai, China). Buffers were prepared by using double distilled water. Calf Thymus DNA (CT-DNA) was purchased from Sigma-Aldrich company and dissolved in the buffers (5 mM Tris (hydroxymethylaminomethane)-HCl, 50 mM NaCl, pH = 7.2). A solution of calf thymus DNA in the buffer gave a ratio of UV absorbance at 260 and 280 nm of ca. 1.8–1.9:1, indicating that the DNA was sufficiently free of protein. The DNA concentration per nucleotide was determined by absorption spectroscopy using the molar absorption coefficient (6600 M^−1^cm^−1^) at 260 nm. 4,6-diamidino-2-phenylindole (DAPI), an ECL-Plus Kit and Cell Cycle and Apoptosis Analysis Kit was purchased from Beyotime (Shanghai, China). 3-(4,5-dimethylthiazol-2-yl)-2,5-diphenyltetrazolium bromide (MTT) was obtained from Sigma-Aldrich. The fluorescent dye 2′,7′-dichlorodihydrofluorescein diacetate (DCF-DA) and 5,5′,6,6′-tetrachloro-1,1′,3,3′-tetraethylbenzimidazolcarbocyanine iodide (JC-1) were purchased from Roche Diagnostics (Indianapolis, IN, USA). Polyclonal antibodies against Bcl-2, Bax and P38 were purchased from Santa Cruz Biotechnology (Santa Cruz, CA, USA). Caspase-3 antibody was purchased from Cell Signaling Technology (Beverly, MA, USA).

### 2.2. Synthesis of Compounds

Binaphthols **1a**–**1c** were obtained according to a method described in previous reports [22,23]. They were oxidated by the complex of CuCl_2_-ethanol amine to afford dibenzoxanthenes **2a**–**2c**. Then, 1 mmol of compounds **2a**–**2c** was dissolved in 20 mL THF. To this, 2 mL of 10% HCl was added and the mixture was stirred for 10 min. The mixture was extracted with ethyl acetate. Then, the obtained extracting solution was irradiated under sunlight for 8 h and 10 mL of 20% NaOH was added to form the corresponding sodium salt. Finally, the solution was acidized by HCl, and a white solid was produced.

3-(12-oxo-12H-benzo[a]xanthen-11-yl)propanoic acid (**3a**): Yield 52%, ^1^H-NMR (500 MHz, DMSO-*d_6_*, Appendix A) *δ*: 12.11 (s, 1H, COOH), 9.82 (d, *J* = 8.0 Hz, 1H, Ar-H), 8.33 (d, *J* = 9.0 Hz, 1H, Ar-H), 8.06 (d, *J* = 9.0 Hz, 1H, Ar-H), 7.79–7.61 (m, 4H, Ar-H), 7.56 (d, *J* = 8.5 Hz, 1H, Ar-H), 7.30 (d, *J* = 7.5 Hz, 1H, Ar-H), 3.55–3.52 (m, 2H, CH_2_), 2.65–2.62 (m, 2H, CH_2_). ^13^C-NMR (125 MHz, DMSO-*d_6_*, Appendix A) *δ*: 179.8, 174.4, 156.5, 156.1, 143.2, 137.4, 134.3, 130.6, 130.4, 129.8, 129.3, 127.5, 126.6, 126.3, 121.3, 118.3, 116.9, 114.9, 35.7, 30.8. HRMS (Appendix A): calcd for C_20_H_15_O_4_: *m/z* = 319.09649, found: *m/z* = 319.09641 (error 0.25 ppm).

3-(3-methoxycarbonyl-12-oxo-12H-benzo[a]xanthen-11-yl)propanoic acid (**3b**): Yield 40%, ^1^H-NMR (500 MHz, Pyr-*d_5_*, Appendix A) *δ*: 10.25 (d, *J* = 9.0 Hz, 1H, Ar-H), 8.71 (s, 1H, Ar-H), 8.41 (dd, *J* = 9.0 Hz 1.5 Hz, 1H, Ar-H), 8.19 (d, *J* = 9.0 Hz, 1H, Ar-H), 7.55–7.51 (m, 2H, Ar-H), 7.38 (d, *J* = 8.0 Hz, 1H, Ar-H), 7.34 (d, *J* = 7.5 Hz, 1H, Ar-H), 4.02–3.99 (m, 2H, CH_2_), 3.82 (s, 3H, OCH_3_), 3.19–3.17 (m, 2H, CH_2_). ^13^C-NMR (125 MHz, Pyr-*d_5_*, Appendix A) *δ**:* 179.6, 166.4, 157.5, 156.1, 149.6, 144.3, 137.3, 133.8, 133.6, 131.1, 129.6, 128.6, 127.5 (2C), 127.0, 121.6, 118.9, 116.5, 115.2, 66.9, 51.9, 31.5. HRMS (Appendix A): calcd for C_22_H_16_NaO_6_: *m/z* = 399.08391, found: *m/z* = 399.08393 (error 0.05 ppm).

3-(3-cyno-12-oxo-12H-benzo[a]xanthen-11-yl)propanoic acid (**3c**) Yield 43%, ^1^H-NMR (500 MHz, Pyr-*d_5_*, Appendix A) *δ*: 10.18 (d, *J* = 9.0 Hz, 1H, Ar-H), 8.29 (s, 1H, Ar-H), 8.11 (d, *J* = 9.0 Hz, 1H, Ar-H), 7.90 (dd, *J* = 9.0 Hz 1.5 Hz, 1H, Ar-H), 7.58–7.55 (m, 2H, Ar-H), 7.41–7.36 (m, 2H, Ar-H), 3.98 (t, *J* = 7.5 Hz, 2H), 3.18–3.15 (m, 2H, CH_2_). ^13^C-NMR (125 MHz, Pyr-*d_5_*, Appendix A) *δ*: 179.4, 175.3, 157.7, 156.0, 144.4, 136.3, 134.1, 133.8, 133.1, 129.9, 129.4, 127.8, 127.7, 121.5, 119.9, 119.1, 116.6, 115.0, 109.4, 36.1, 31.4. HRMS (Appendix A): calcd for C_21_H_13_NNaO_4_: *m/z* = 366.07368, found: *m/z* = 366.07362 (error 0.16 ppm).

### 2.3. Cell Culture

Roswell Park Memorial Institute Medium (RPMI-1640) was used to culture BEL-7402 and SGC-7901 cell lines; and A549, HeLa and LO2 cells were cultured in Dulbecco’s Modified Eagle’s Medium (DMEM) containing 10% fetal bovine serum (FBS). Cell passage experiments were performed with trypsin every 2 days to maintain exponential growth. All cells were cultured until they reached logarithmic growth phase, unless specifically noted.

### 2.4. In Vitro Cytotoxicity Assay

The cytotoxicity test was carried out according to a method described in the literature [24]. The cells were seeded into 96-well micro-assay culture plates and grown overnight in a 5% CO_2_ incubator at 37 °C. Then, the cells were treated with gradient concentrations of the compounds when the cells grew to 60–70%. At 48 h posttreatment, the medium was removed and MTT solution was added. After 4 h, the absorbance at 490 nm was measured with a microplate reader. The IC_50_ values were calculated by SPSS (Statistical Product and Service Solutions) software.

### 2.5. Partition Coefficients

Partition coefficients of compounds in octanol and phosphate buffer (0.129 M NaCl, pH = 7.4) were measured as described by Owellen and coworkers [25]. The concentrations of the compounds in the octanol and phosphate buffer phases were determined three times using a UV/Vis spectrophotometer. The partition coefficients were calculated from the ratio C_oct_/C_PBS_, where C_oct_ and C_PBS_ were the concentrations of the compound in the octanol and phosphate buffer, respectively.

### 2.6. Determination of pK_a_ Values

The p*K_a_* values of the compounds were determined by potentiometric titration using a pH meter (Basic pH Meter PB-10, Sartorius) calibrated with standard buffers of pH 4.01, 6.86 and 9.18 [26]. The compounds were dissolved in aqueous dimethyl sulfoxide (30% dimethyl sulfoxide) at constant ionic strength (0.15 M KCl). The solution was pre-acidified to approximately pH = 2 with 0.24 M HCl, and titrated until pH = 12 with 0.06 M KOH.

### 2.7. DNA Binding Experiments

#### 2.7.1. Electronic Absorption Titration

Calf thymus DNA (CT-DNA) was dissolved in 5 mM Tris-HCl buffers and stored at 4 °C. The DNA concentration was measured by UV absorbance of 260 nm through the following Equation (1) [27]:[DNA] = K·A_260_/6600(1)
where K is dilution ratio, A_260_ stands for the absorbance value of DNA at 260 nm. The electronic absorption spectra were recorded from 200 to 400 nm and the DNA binding constants (K_b_) were calculated through the following Equation (2) [28]:[DNA]/(ε_a_ − ε_f_) = [DNA]/(ε_b_ − ε_f_) +1/K_b_ (ε_b_ − ε_f_)(2)
where the [DNA] was the concentration of DNA in the base pairs. ε_a_, ε_b_ and ε_f_ represented the apparent absorption coefficient A_obsd_/compound, the extinction coefficient of the compounds with fully bound and free forms, respectively. In plots of [DNA]/(ε_a_ − ε_f_) versus [DNA], K_b_ was given by the ratio of the slope to the intercept.

#### 2.7.2. Luminescence Spectra Titration

We used ethidium bromide (EB)-competing binding experiments to investigate the binding mode between the compounds and CT-DNA. It is well known that EB combines DNA molecules through a classic intercalative mode. Maintaining the DNA-EB solution concentration (DNA:EB = 1.25:1) and varying the concentrations of compounds (6.62–50 μM), the spectra were recorded after 5 min. The Stern-Volmer constants (K_sv_) were calculated using the following Equation (3) [29]:*F*_0_/*F* = 1 + K_sv_ [Q](3)
where *F*_0_ and *F* were the fluorescence intensities of the EB-CT DNA solution at 550–700 nm in the absence and presence of the compounds, respectively. [Q] was the concentration of compounds. In plots of *F*_0_*/F* versus [Q], K_sv_ was given by the ratio of the slope to the intercept.

#### 2.7.3. Viscosity Experiments

An ubberhold viscometer was used to measure the viscosity of the CT-DNA solution in the absence and presence of compounds, and the temperature was maintained at 25 ± 0.1 °C. The concentration of CT-DNA was 200 μM, and the concentrations of the compounds varied from 0 to 90 μM. The flow times were recorded with a digital stopwatch; each sample was measured three times to get the average time. The results were demonstrated as (*η*/*η*_0_)^1/3^ versus r (r = [Compound]/[DNA]), where *η* and *η*_0_ were the relative viscosity of CT-DNA solutions in the presence and absence of compounds, and the viscosity was calculated by the following equation [30]:*η* = (*t* − *t**_o_*)/*t**_o_*(4)
where *t* is the flow time of the DNA-compound solution and *t_o_* is the flow time of Tris buffer.

### 2.8. Molecular Docking Studies

The geometry structures of complexes were optimized using the DFT-B3LYP method with the 6-31G* basis set for the C, N, O, H atoms with the GAUSSIAN-09 quantum chemistry program-package [31]. Molecular docking was performed with the AutoDock 4.2 Lamarckian Genetic Algorithm (LGA) [32,33]. The dsDNA structure was obtained from the protein data bank (PDB ID: 4E7Y).

### 2.9. Cellular Uptake Studies

Cellular uptake studies were performed according to a method described in the literature [34]. SGC-7901 cells were placed in 24-well microassay culture plates and grown overnight at 37 in a 5% CO_2_ incubator. The compounds tested were then added to the wells. The plates were incubated for 48 h. Upon completion of incubation, the wells were washed three times with phosphate buffered saline (PBS). After moving the culture medium, the cells were stained with 2-(4-amidinophenyl)-6-indolecarbamidine dihydrochloride (DAPI) and observed under fluorescence microscope.

### 2.10. Cell Cycle Arrest

The cell cycle arrest was carried out using a method described in the literature [35]. Briefly, SGC-7901 cells were treated with IC_50_ concentrations of compounds for 24 h. Then, the cells were harvested and fixed overnight in 70% ethanol at 4 °C. Next, the permeabilized cells were stained with PI/RNase buffer and the DNA content was determined on a flow cytometer. The results were analyzed by ModFit LT 4.0 (Verity Software House, Topsham, ME, USA).

### 2.11. Apoptosis Assays by AO/EB Staining and Flow Cytometry

SGC-7901 cells were seeded onto chamber slides in six-well plates at a density of 1.2 × 10^5^ cells per well and were incubated for 24 h. Then, the medium was removed and replaced with medium containing the compounds. After 24 h, the solution was removed and the cells were washed three times with ice PBS. Next, the cells were stained with acridine orange (AO) and ethidium bromide (EB) (AO: 100 μg mL^−1^, EB: 100 μg mL^−1^) solution for 15 min at 37 °C. The cells were observed and imaged under ImageXpress Micro XLS system (MD company, San Jose, CA, USA). We also used flow cytometry to determine the apoptosis of GC-7901 cells. Cells were seeded in a 12-well plate and incubated overnight. Then, the cells were treated with compounds for 24 h. After treatment, the cells were harvested, washed three times with ice-cold PBS, and then resuspended in 195 μL Annexin V-FITC binding buffer. Next, 10 μL PI and 5 μL Annexin-FITC were added to each sample, the sample was stained in the dark for 20 min, and was analyzed by a FACS Calibur flow cytometer (Beckman Dickinson & Co., Franklin Lakes, NJ, USA) [36].

### 2.12. Measurement of Reactive Oxygen Species

SGC-7901 cells were seeded in 12-well plate at a density of 1.5 × 10^5^ cells per well for 24 h. Then, the medium was replaced with medium containing compounds for 24 h. After treatment, the medium was removed and washed three times with PBS, and the cells were stained with 20 μM DCFH-DA for 30 min in the dark. Finally, the cells were imaged under ImageXpress Mico XLS system (MD company, San Jose, CA, USA) [37].

### 2.13. Assays of Location of the Compounds at the Mitochondria

SGC-7901 cells were plated in a 12-well plate at a density of 1.5 × 10^5^ cells per well overnight. The medium was removed and replaced by different concentrations of the compounds for 24 h at 37 °C. Then, the plate was washed three times with PBS to remove residual compounds, and the cells were further incubated with Mito-tracker Red for 30 min. After that, the cells were washed twice with ice-cold PBS to remove residual compounds and imaged under ImageXpress Mico XLS system (MD company, San Jose, CA, USA) [38].

### 2.14. Intracellular Malondialdehyde (MDA) and GSH Assay

After incubation with IC_50_ concentrations of **3b** and **3c** for 24 h, SGC-7901cells were lysed. The lysate was centrifuged for 15 min to collect the supernatant. The protein concentration was determined using a BCA Protein Assay Kit (Beyotime, Shanghai, China). A lipid peroxidation MDA assay kit (Beyotime, Shanghai, China) was used to detect MDA content, and a GSH and GSSG Assay Kit was used to detect intracellular GSH. SGC-7901 cells were incubated with IC_50_ concentrations of **3b** and **3c** for 24 h. Afterwards, the cells were lysed by the freeze-thaw method, and the lysate was centrifuged for 5 min to collect the supernatant. The absorbance was measured at 412 nm with a microplate reader [39].

### 2.15. Mitochondrial Membrane Potential Assay (∆Ψm)

SGC-7901 cells were treated with IC_50_ concentrations of the compounds in 12-well plates for 24 h and were washed three times with cold PBS. Then, the cells were stained with 1 μg/mL JC-1 for 30 min at 37 °C in the dark. After that, the cells were washed twice with ice-cold PBS to remove residual stain and imaged under the ImageXpress Micro XLS system (MD company, San Jose, CA, USA) [40].

### 2.16. Immunostaining Analysis

The SGC-7901 cells were cultured in 12-well plates and treated with IC_50_ concentrations of **3b** and **3c** for 24 h. The cells were fixed with 75% alcohol, and then washed and incubated with immunostaining blocker for 1 h. The primary antibody CRT Antibody, HSP70 Antibody and HMGB1 antibody were incubated for 12 h and then the cells were washed three times with PBS. Next, the secondary antibody FITC-labeled Goat Anti-Rabbit IgG was incubated in darkness for 1 h. The cells were washed three times with PBS. Finally, the nuclei were stained with Hoechst and photographed under Micro XLS System (MD company, San Jose, CA, USA) [41].

### 2.17. Intracellular ATP Measurement

The SGC-7901 cells were cultured in six-well plates and treated with IC_50_ concentrations of **3b** and **3c** for 24 h. The cells were lysed for 10 min and centrifuged to obtain the supernatant. The prepared ATP detection working solution was added into the supernatant, 5 min late, the RLU value was measured [42].

### 2.18. Determination of Intracellular Ca^2+^ Levels

Fluo-3AM, a Ca^2+^-sensitive fluorescent probe, was used to detect the level of intracellular Ca^2+^. SGC-7901 cells were incubated with IC_50_ concentrations of the compounds for 24 h at 37 °C. Then, the cells were washed three times with PBS, incubated with Fluo-3AM, and observed under an ImageXpress Micro XLS system (MD company, San Jose, CA, USA). The fluorescence intensity was analyzed under a high content analysis system [43].

### 2.19. Western Blot Analyses

SGC-7901 cells were seeded in six-well plates for 24 h and incubated with IC_50_ concentrations of the compounds in the presence of 10% fetal bovine serum (FBS). The cells were then harvested in lysis buffer. After sonication, the samples were centrifuged for 20 min at 13,000× *g*. The protein concentration of the supernatant was determined by BCA (bicinchoninic acid) assay. Sodium dodecyl sulfate–polyacrylamide gel electrophoresis was carried out with loading equal amounts of proteins per lane. The gels were then transferred to poly (vinylidene difluoride) membranes (Millipore, Billerica, MA, USA) and blocked with 5% nonfat milk in TBST (20 mM Tris-HCl, 150 mM NaCl, 0.05% Tween 20, pH 8.0, Tween: polyoxyethylene monolaurate sorbaitan) buffer for 3 h. The polyvinylidene difluoride membranes were washed four times with TBST (4 × 10 min) and incubated with primary antibody solution at 4 °C overnight. They were then washed four times with TBST for a total of 30 min. The secondary antibodies were then conjugated with horseradish peroxidase (1:5000 dilution) for 70 min at room temperature and washed four times with TBST. The blots were visualized with the Amersham ECL (electrochemiluminescence) and western blotting detection reagents according to the manufacturer’s instructions. To assess the presence of comparable amounts of proteins in each lane, the membranes were stripped finally to detect the β-actin [44].

### 2.20. Statistical Analysis

All data are expressed as the mean ± SD. Differences between two groups were analyzed by a two-tailed Student’s test. Differences with * *p* < 0.05 are considered statistically significant.

## 3. Results and Discussion

### 3.1. Chemistry and Stability Studies

The method of synthesis of benzoxanthones derivatives **3a**–**3c** is shown in Figure 2. According to a method described in a previous report [22,23], the oxidation of binaphthols **1a**–**1c** provided dibenzoxanthenes **2a**–**2c**. In acidic conditions, benzoxanthones derivatives **3a**–**3c** were synthesized through nucleophilic substitution, proton migration and ring-opening reaction of compounds **2a**–**2c** under irradiation. The structures of benzoxanthones **3a**–**3c** were characterized by NMR and HRMS. In the spectra of HRMS, the determined molecular weights were consistent with the expected values. In the ^13^C NMR spectra, the chemical shifts of 179.8 ppm for **3a**, 179.6 ppm for **3b** and 179.4 ppm for **3c** were attributed the presence of carbonyl groups. The peaks of 174.4, 166.4 and 175.3 ppm were assigned to the -COOH for **3a**, **3b** and **3c**, respectively, whereas the chemical shifts of 35.7 ppm for **3a**, 51.9 ppm for **3b**, 36.1 ppm for **3c** were attributed to the carbon at the 1-position, 30.8 ppm for **3a**, 31.5 ppm for **3b**, 31.4 ppm for **3c** are assigned to the carbon at the 2-position. We further speculated on the reaction mechanism for **3a** (Figure 3): Firstly, 2a → i was performed according to the method described in our previous report [45]. Then, it was easy for chlorine atoms to leave and be replaced by OH^–^ to form ii. Generally, photochemical reactions take place through free radical mechanisms [46]. Hence, ii → iii was carried out through the photochemical reaction to produce free radicals, followed by hydrogen rearrangement to obtain ketene (iv). Subsequently, compound (v) was formed by the nucleophilic addition of ketene (iv) with H_2_O. Finally, the keto enol tautomerism reaction of the compound (v) generated product **3a**.

The stability of **3b** and **3c** in PBS was studied by UV-Vis spectra. As shown in Figure 1, at 0 and 48 h, the shapes of the peaks were the same, which indicated that **3b** and **3c** were stable in PBS. Owing to low solubility in PBS, at 48 h, some precipitate was observed. Hence, there was a difference in the absorbance.

### 3.2. The Cytotoxicity, Partition Coefficient and pKa Value of Compounds

The acid-base properties of compounds are characterized by an acid dissociation constant (p*K_a_*). This physicochemical property is an important parameter because it affects molecule biological, metabolism, toxicity, cellular uptake or extrusion, tissue distribution and so on [47,48]. We detected the p*K_a_* values of compounds **3b** and **3c** using potentiometric titration. The p*K_a_* values were obtained by a graph of pH versus volume of KOH. As shown in Figure 2, the p*K_a_* values were 7.48 for **3b** and 7.39 for **3c**, respectively. The compounds with at least one charge with a p*K_a_* < 4 for acids and a p*K_a_* > 10 for bases do not cross the blood-brain barrier by passive diffusion [49]. The p*K_a_* values of **3b** and **3c** indicated that the compounds could enter the cells.

The antiproliferation of benzoxanthones **3a**–**3c** against SGC-7901, A549, HeLa, BEL-7402 and LO2 was evaluated by the MTT method. The obtained data are expressed as IC_50_ (50% inhibitory concentration) values. As shown in Table 1, compound **3a** showed no cytotoxic activity toward the selected cell lines with IC_50_ values more than 100 µM, while compounds **3b** and **3c** with electron-withdrawing group exerted antiproliferative effects on the selected tumor cell lines except BEL-7402 cells. Comparing the cytotoxic activity, compounds **3b** and **3c** showed the highest cytotoxic effect on SGC-7901 among these cancer cells. In particular, **3c** exhibited higher cytotoxicity than **3b** toward all selected cancer cells. In our previous report, the compounds with cyano groups showed high antitumor activity [50]. We consider that the strong electron-withdrawing cyano group can increase the cytotoxicity of compound. This is consistent with the results obtained from the DNA-binding affinities of **3c** > **3b**.

It is well known that the lipophilicity of drugs is an important contributor to cytotoxicity [51]. Lipophilicity is evaluated by the partition coefficient in octal/water systems. The partition coefficient can provide valuable information to understand the uptake, distribution, biotransformation and degradation of drugs [52]. To further analyze the cytotoxic activity of **3c** > **3b**, we measured the partition coefficient of the compounds by shake flask method. The log*P* of two compounds is shown in Table 1. The value of **3c** was larger than that of compound **3b,** indicating **that** it is easy for **3c** to enter the cells compared with **3b**.

### 3.3. Cellular Uptake Studies

The *pK_a_* values and partition coefficient show that the compounds can enter the cells. The cell uptake was studied using DAPI (2-(4-amidinophenyl)-6-indolecarbamidine dihydrochloride) as the fluorescence probe. As shown in Figure 3, after 24 h treatment of SGC-7901 cells with IC_50_ concentrations of **3b** and **3c**, the cell nuclei stained by DAPI showed blue, the compounds emitted green fluorescence, and the merge represented the cell binding of the compounds. The overlap of blue and green fluorescence indicated that the compounds could be successfully endocytosed by the cells and the compounds could enter the cytoplasm and accumulate in the nuclei.

### 3.4. Cell Cycle Arrest

Antitumor efficacy is closely related to the cell cycle. Numerous drugs have an impact on cycling cells. The cell cycle progression induced by the compounds was measured using flow cytometry. As shown in Figure 4, the cells treated with **3b** and **3c** accumulated at G0/G1 phase for 74.6% and 76.7% with a decrease at the G2/M phase for 17.4% and 16.0%. An increase of 17.7% for 3b and 19.8% for 3c in the cell at G0/G1 phase was discovered. The results showed that compounds induced cell cycle arrest at G0/G1 phase, namely, the compounds inhibited the cell proliferation at G0/G1 phase.

### 3.5. Apoptosis Studies with AO/EB Double Staining and Flow Cytometry

Acridine orange and ethidium bromide double staining assay was used to explore the cell morphological changes (such as cell shrinkage, chromatin condensation, nuclear membrane blebbing). As shown in Figure 5a, the cells in the control group appeared clear and intact in terms of the cell nuclei structure; however, after treatment of SGC-7901 cells with IC_50_ concentrations of the compounds for 24 h, apoptotic features such as cell shrinkage and chromatin condensation were observed. This shows that the compounds could induce apoptosis. To further investigate the apoptotic effect of compounds on SGC-7901 cells, Annex V/PI staining was applied to quantify the percentage of apoptosis cells. As shown in Figure 5b, compared with control group, the percentage of early apoptosis cells increased to 6.68% for **3b** and 8.65% for **3c**, respectively. These data further demonstrate that the compounds can effectively cause apoptosis in SGC-7901 cells.

### 3.6. Intracellular Oxidative Stress Increase Detection

Apoptosis-associated factor reactive oxygen species (ROS) play a central role in the regulation of cellular apoptosis. It can activate a series of apoptotic signaling pathways as an upstream factor [53]. Excessive ROS production is relevant to biological macromolecule damage and the early stages of apoptosis [54]. Therefore, we used DCFH-DA as a ROS probe to detect the change of intracellular ROS levels. ROS produced within the cells oxidize DCFH to the highly fluorescent compound, namely, 2′,7′-dichlorofluorescein (DCF). As shown in Figure 6a, after treatment with Rosup (positive control), IC_50_ concentrations of **3b** and **3c** for 24 h, an obvious increase of DCF fluorescence in SGC-7901 cells was observed compared with control cells. This suggested that excessive ROS was accumulated. To quantitatively compare the effects of compounds on ROS levels, the DCF fluorescence intensity was determined by high content analysis. As shown in Figure 6b, the fluorescence intensity of DCF increased by about 60.5% for **3b** and 93.1% for **3c**.

P38 MAPK, as a vital part of MAPK pathway, participates in a signaling cascade controlling cellular responses to cytokines and stress. It can be activated by a range of cellular stresses such as inflammatory cytokines, ultraviolet light and growth factors. Furthermore, oxidative stress can indirectly activate P38 and further induce apoptosis [55,56,57]. To prove whether the studied compounds induced apoptosis through a ROS-P38 pathway, we detected P38 MAPK protein expression by western blot. As shown in Figure 6c, after treatment with IC_50_ concentrations of compounds **3b** and **3c** for 24 h, the P38 MAPK expression was significantly upregulated, suggesting that the compounds could cause oxidative stress and further induce apoptosis through the ROS-P38 pathway.

### 3.7. Intracellular MDA and GSH Detection

A large amount of ROS can produce lipid oxidation of mitochondrial membrane [58]. Malondialdehyde (MDA) is a natural product of lipid oxidation. Compounds **3b** and **3c** were shown to enhance intracellular ROS in cells; therefore, we further detected the MDA level in SGC-7901 cells. As shown in Figure 7a, compared to control cells, the cells treated with IC_50_ concentrations of compounds **3b** and **3c** showed a significant increase in MDA levels. The increase of MDA content indicated the occurrence of oxidative stress reaction; MDA can cause the cross-linking polymerization of protein and nucleic acid. The cytotoxic lipid oxidation end product MDA affects mitochondrial respiration and disrupts mitochondrial redox ability, further disrupting the intracellular environment. Glutathione (GSH) is an important antioxidant in terms of maintaining homeostasis and resisting oxidative stress. GSH binds with peroxides and free radicals to induce oxidative damage [59]. As shown in Figure 7b, a marked decline in GSH concentration was found compared with the control after the treatment of SGC-7901 cells with IC_50_ concentrations of compounds. In summary, compounds **3b** and **3c** inhibit the generation of GSH in cells, reducing the antioxidant capacity of cells, increasing ROS content, causing oxidative stress reaction and inhibiting cell growth.

### 3.8. Mitochondrial Membrane Damage Detection

ROS is mostly derived from the metabolism of mitochondria. It is reasonable to hypothesize that compounds act on mitochondria and cause mitochondrial dysfunction. Mitochondrion is a crucial subcellular organelle that plays an important role in cell death, neoplasia, cell differentiation, oxygen and hypoxia sensing, and calcium metabolism [60]. Damage to the mitochondrial membrane induces a series of reactions including mitochondrial membrane potential reduction and cytoplasmic Ca^2+^ increase [61]. As illustrated in Figure 8a, after SGC-7901 cells were treated with IC_50_ concentrations of the compounds for 24 h, the mitochondria were stained red and the compounds emitted green fluorescence. The merge of the red and green fluorescence suggested that the compounds were located in the mitochondria. This result indicates that mitochondria were the target of compounds. A JC-1 dye assay was used to evaluate whether the compounds could cause mitochondrial membrane damage according to the change from red (JC-1 aggregated form in mitochondria at high membrane potential) to green (JC-1 monomeric form in cytosol at low membrane potential) [62]. As shown in Figure 8b, after exposure of SGC-7901 cells to the IC_50_ concentrations of compounds **3b** and **3c** for 24 h, we found a decrease of red fluorescence and an increase of green fluorescence. This change confirmed that mitochondrial membrane potential had decreased. The ratio of red/green fluorescence was 5.85 in control. The ratio of red/green fluorescence was 3.01 for **3b** and 0.10 for **3c**. A JC-1 staining assay showed that compounds were the cause of the mitochondria dysfunction.

### 3.9. Immunogenic Cell Death (ICD) Induced by Compounds

Immunogenic cell death (ICD) is a form of cell death in which ICD occurs in tumor cells; a series of signaling molecules are involved, including the release of calreticulin (CRT), heat-shock protein (HSP70), adenosine triphosphate (ATP) and high mobility group box 1 (HMGB1) [41]. ICD is believed to overcome drug resistance caused by tumor, enhance antitumor immunity, and achieve better therapeutic effect [63]. In order to explore the potential of the studied compounds to induce ICD, we examined CRT and HSP70, HMGB1 and ATP release, i.e., the four key markers for ICD. As shown in Figure 9, we found that **3b** and **3c** increased the expression of CRT, HSP70 and HMGB1 in tumor cells. Compared with the control group, the green fluorescence in cytoplasm of all three groups was increased, and the green fluorescence of HMGB1 was obviously transferred from the original nuclei to the cytoplasm. Treatment of SGC-7901 cells with IC_50_ concentrations of **3b** or **3c** showed a significant increase in ATP concentration. The increase of HMGB1, HSP70 and CRT in cytoplasm, along with the increase of ATP content indicated that the compounds induced an increase of ICD. Therefore, we consider that **3b** and **3c** can cause oxidative stress in cells and eventually lead to immunogenic death.

### 3.10. Mechanism Studies of the Mitochondrial Apoptotic Pathway

The mitochondrial pathway is the most prevalent mechanism of apoptosis. More importantly, in the early stage of apoptosis, the phenomenon of endometrial permeability changes, leading to calcium ion uptake, membrane potential decrease, cytochrome C release and release of apoptosis-inducing factors [64]. To confirm that the compounds induced apoptosis through the mitochondrial pathway in SGC-7901 cells, the expression level of Ca^2+^, cytochrome C and some apoptosis-inducing factors was detected and quantified. Ca^2+^ overload is associated with mitochondria dysfunction and cell apoptosis [65]. SGC-7901 cells were stained with the Ca^2+^ sensitive dye Flu-3-AM and the level of Ca^2+^ was assessed using an ImageXpress Micro XLS system. As depicted in Figure 10a, after treatment with IC_50_ concentrations of compounds **3b** and **3c**, a number of bright green fluorescence images in SGC-7901 cells appeared; in contrast, those in the control group appeared with weak fluorescent intensity. The results show that the compounds can increase intracellular Ca^2+^ levels. To quantitatively detect the fluorescence intensity of the intracellular Ca^2+^, high content analysis was used. As shown in Figure 10b, the fluorescence intensity of Flu-3-AM was 66.09 × 10^5^ in the control. Nevertheless, the fluorescence intensity was increased by 1.67 times for **3b**, 1.36 times for **3c**, indicating that the compounds induced an increase of Ca^2+^ levels. On the other hand, the Cyt c release is a major trigger of cascade activation and has been treated as a highly specific event in apoptotic signaling [66]. Cyt c was measured using an ImageXpress Micro XLS system and high content analysis. As shown in the Figure 10c, no obvious fluorescent spots were observed in the control group. However, after treatment with IC_50_ concentrations of compounds for 24 h, the fluorescence intensity was significantly increased. Compared to the control (Figure 10d), the fluorescence intensity was increased by 1.41 times for **3b** and **3c**, revealing the Cyt c was released from mitochondria and concentrated in cytoplasm. In order to explore the mechanism of apoptosis induction, the expression of PARP, caspase 3 and Bcl-2 family proteins was detected by western blot assay. As shown in Figure 10e, after treatment with IC_50_ concentrations of compounds, the expression of PARP and caspase 3 increased, indicating that PARP and caspase 3 had been activated. Bcl-2 family protein is crucial factors of the mitochondrial apoptotic pathway. The Bax/Bcl-2 ratio in the cells can regulate the susceptibility of cells to apoptosis. As seen in Figure 10e, Bcl-2 expression was downregulated and Bax expression was upregulated, and the ratio of Bax/Bcl-2 was increased, which further demonstrates that the compounds induced apoptosis.

### 3.11. The DNA Binding Studies of Compounds

Many drugs, such as cisplatin, possess anticancer activity in terms of their DNA binding affinity. Planar aromatic ring compounds could intercalate DNA molecules by means of their planar structure, overlapping with DNA base pairs and stabilized by hydrogen bonds, Van der Waals forces and π-π stacking [67]. Due to the existence of intercalative interactions between the compounds and DNA, the base pairs of DNA are separated and the helix is untwisted, which makes it impossible or difficult to replicate; thus, compounds can exert antitumor and antiviral activity [68]. In this paper, we investigated the interaction of compounds **3b** and **3c** with DNA through electronic absorption titration. The absorption spectra of compounds **3b** and **3c** are depicted in Figure 11a in the presence and absence of CT-DNA. It was clearly seen that hypochromism was observed and the spectra exhibited slight red shift. To compare the DNA binding ability of these compounds, the intrinsic binding constant K_b_ was determined. The K_b_ values were 1.64 (±0.28) × 10^4^ M^−1^ for **3b**, and 3.55 (±0.41) × 10^4^ M^−1^ for **3c**. The DNA-binding intensity of the compounds followed the order of **3c** > **3b**.

To further investigate the interaction mode between compounds and DNA, fluorescence titrations were performed. An EB-competitive binding experiment was used to investigate the interaction between compounds and CT-DNA. It is well known that ethidium bromide (EB) can combine DNA molecule through a classic intercalative model. When compounds compete with EB, the EB is released from the DNA double helix, and the fluorescence intensity is significantly reduced. As shown in Figure 11b, by increasing the concentrations of the compounds, the fluorescent intensity of EB-DNA gradually decreased, indicating that the compounds interact with CT-DNA through competition with EB. This also verifies that the compounds interact with CT-DNA via the intercalative mode. The Stern-Volmer constants (K_sv_) were obtained using Equation (3); the K_sv_ values were 2.20 (±0.12) × 10^3^ M^−1^ for **3b** and 3.28 (±0.21) × 10^3^ M^−1^ for **3c**, respectively. According to above results, we consider that two compounds interact with CT-DNA via a classic intercalative mode.

Viscosity experiments are used to detect the mode of interactive between small molecules and DNA. The results can provide more information compared with spectral data. When small molecule compounds interact with DNA in a nonintercalative mode such as groove and electrostatic binders, the viscosity of DNA solution has negligible change. In the case of the classic intercalation, the distance between adjacent base pairs increases due to the intercalation of small molecules, resulting in the elongation of the double helix, which increase the viscosity of the DNA solution; the greater the binding strength, the greater the change in viscosity. In contrast, the nonclassic or partial intercalation will cause the double helix to become kinked, resulting in a decrease of viscosity [69]. As shown in Figure 11c, with increasing the compound concentrations, the relative viscosities of CT-DNA increased. Moreover, in the presence of compound **3c**, the viscosity of CT-DNA underwent a larger change than that of compound **3b**, which further shows that the compounds interact with DNA through classic intercalative mode.

In recent years, molecular docking analysis has been widely applied to elucidate the essence of the interactions between compounds and DNA [70]. The structures of compounds were geometrically optimized. As shown in Figure 11d, the molecular docking showed that the aromatic-ring plane of **3b** and **3c** intercalated into the DNA base pairs. This further demonstrated that compounds **3b** and **3c** interacted with DNA through intercalative mode. The above DNA-binding results suggest that the compounds may target the DNA molecules.

Taken together, we consider that the benzoxanthone analogues induce cell apoptosis through two main pathways (Figure 12). (I) Mitochondrial apoptosis pathway: The compounds target mitochondria and damage mitochondrial membranes, increase the intracellular MDA and GSH, enhance the levels of ROS and subsequently affect the release of Ca^2+^. Excessive Ca^2+^ accelerates the damage of mitochondria, which releases the cytochrome C to activate PARP, caspase 3 and induces apoptosis. Compounds can also enhance the expression of HMGB1, HSP70 and CRT in cytoplasm, along with increasing the ATP content to induce ICD. (II) Cell cycle arrest pathway: The analogues can bind to CT-DNA through intercalative mode and damage the normal structure of DNA, which makes it difficult for the DNA to replicate. These factors trigger cell cycle arrest and further induce cell apoptosis.

## 4. Conclusions

Three new benzoxanthones have been prepared through photochemical synthesis, and their structures have been characterized by NMR and HRMS. Compounds **3b** and **3c** exhibited obvious antiproliferation against SGC-7901 cells. An AO/EB staining assay indicated that compounds induced the apoptosis of tumor cells. Subcellular localization suggested that compounds could enter mitochondria, resulting in the loss of mitochondrial membrane potential and the release of cytochrome C. Compounds **3a** and **3b** also inhibited the generation of GSH, increased ROS content, caused oxidative stress in cells and eventually led to immunogenic death. The compounds could upregulate the expression of PARP, Bax, and downregulate the expression of Bcl-2 protein. In addition, the compounds combined with CT-DNA by intercalative mode and induced cell cycle arrest at G0/G1 phase. Taken together, we consider that these compounds induce apoptosis through a ROS-mediated mitochondrial dysfunction pathway. Additionally, the compounds intercalate between DNA base pairs, which causes a change of DNA structure, allowing the compounds inhibit cancer cell proliferation. Our work is helpful for designing and synthesizing new xanthene compounds as potential anticancer candidate drugs and understanding the molecular mechanisms of apoptosis.

## Data Availability

All data are available in the manuscript and Appendix A.

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
