# Peer review of "Synthesis, Characterization and Anticancer Efficacy Evaluation of Benzoxanthone Compounds toward Gastric Cancer SGC-7901"

_molecules, 2022, doi:10.3390/molecules27061970_

Round 1
Reviewer 1 Report
In my opinion, the manuscript needs a significant revision, especially the synthesis additions.
- The authors describe the synthesis of benzoxanthones and very interesting research on their biological properties. However, in my opinion, the preparation of these compounds is not properly documented. The starting materials used are the binaphthol derivatives (1a-1c) from which, after a series of transformations, the products (3a-3c) are obtained as shown in Scheme 1.
The synthesis of compounds 2b and 2c and their spectroscopic data are described in the cited ref. 20. However, there is no compound 2a described there - needed for the synthesis of the product 3a. I did not find it in the prepared manuscript either.
- The authors proposed a complicated, sequential mechanism for the transformation of 2a to 3a. However, they did not mention whether they had any evidence of the proposed mechanism. Did they manage to catch/observe and analyze any of the proposed intermediates (i, ii, iii, iv or v, see Scheme 2.) in any way? So what do they base their proposition on?
There was no cited literature that could describe a similar or analogous process as the basis for the proposed one.
- There is a lack of Supplementary Materials containing spectroscopic spectra of the products.
Author Response
1. The authors describe the synthesis of benzoxanthones and very interesting research on their biological properties. However, in my opinion, the preparation of these compounds is not properly documented. The starting materials used are the binaphthol derivatives (1a-1c) from which, after a series of transformations, the products (3a-3c) are obtained as shown in Scheme 1.
The references 22 and 23 for 1a-1c have been added in the “synthesis of compounds”.
2. The synthesis of compounds 2b and 2c and their spectroscopic data are described in the cited ref. 20. However, there is no compound 2a described there - needed for the synthesis of the product 3a. I did not find it in the prepared manuscript either.
The spectroscopic data of compound 2a can be observed in ref 23.
3. The authors proposed a complicated, sequential mechanism for the transformation of 2a to 3a. However, they did not mention whether they had any evidence of the proposed mechanism. Did they manage to catch/observe and analyze any of the proposed intermediates (i, ii, iii, iv or v, see Scheme 3.) in any way? So what do they base their proposition on?
We have added reference to explain the reaction mechanism and the mechanism has been rewritten.
4. There was no cited literature that could describe a similar or analogous process as the basis for the proposed one.
We have provided the ref 45 and 46 for the mechanism of the reaction.
5. There is a lack of Supplementary Materials containing spectroscopic spectra of the products.
We have provided HRMS, 1HNMR and 13CNMR spectra of the products in the supplementary materials.

Reviewer 2 Report
1. Introduction and discussion should be focused more around the observations and novelty of this study with supported with related references. The authors may be use the following references.
a. https://doi.org/10.3390/molecules26195827
b. https://doi.org/10.1007/s00217-020-03515-x
c. https://doi.org/10.1016/j.molstruc.2020.128588
d. DOI: 10.3390/ph14060509
2. Many typos mistakes were observed , Thus, English language editing is required.
3. Methods can be supported with related references. References must be written according to journal style.
4. More concluding remarks must be also added.
Author Response
1. Introduction and discussion should be focused more around the observations and novelty of this study with supported with related references. The authors may be use the following references.
- https://doi.org/10.3390/molecules26195827
- https://doi.org/10.1007/s00217-020-03515-x
- https://doi.org/10.1016/j.molstruc.2020.128588
- DOI: 10.3390/ph14060509
The above references have been cited as Ref 7,8, 24, 36
2. Many typos mistakes were observed, Thus, English language editing is required.
We have carefully checked the manuscript and some typos mistakes have been revised.
3. Methods can be supported with related references. References must be written according to journal style.
we have added relevant references for methods.
4. More concluding remarks must be also added.
We have added more concluding remarks in conclusions.

Reviewer 3 Report
Prof. Wang and his coworkers have succeeded in the synhteisis of three benzoxanthone derivatives using photochemical reactions. Various analytical methods were used to characterize those compounds. Compounds 3b and 3c were found to have anticancer activity, indicating that these derivatives are expected to be candidates for anticancer drugs in the future.
My evaluation is that the paper is publishable with minor scientific revisions but with substantial language revisions.
- On page 8, second paragraph, is there a citation for the presence of a strong electron-withdrawing group to enhance cyto-toxicity? Or do you have results that show strong activity with the introduction of a nitro group?
- There were a few spelling and citation errors. Please revise to the confirmed pdf attached and correct them.
Author Response
Prof. Wang and his coworkers have succeeded in the synthesis of three benzoxanthone derivatives using photochemical reactions. Various analytical methods were used to characterize those compounds. Compounds 3b and 3c were found to have anticancer activity, indicating that these derivatives are expected to be candidates for anticancer drugs in the future.
My evaluation is that the paper is publishable with minor scientific revisions but with substantial language revisions.
1. On page 8, second paragraph, is there a citation for the presence of a strong electron-withdrawing group to enhance cyto-toxicity? Or do you have results that show strong activity with the introduction of a nitro group?
We provided the reference [50] to explain for the presence of electron-withdrawing group to enhance cyto-toxicity.
2. There were a few spelling and citation errors. Please revise to the confirmed pdf attached and correct them.
We have revised a few spelling and citation errors.

Reviewer 4 Report
The manuscript “Synthesis, characterization and anticancer efficacy evaluation 1 of benzoxanthone compounds toward gastric cancer SGC-7901” describes investigations on anticancer action of a series of novel benzoxantone derivatives. Biological part of the paper is performed in a standard for the authors protocol, and I am not experienced enough to judge it. However, the manuscript contains chemical and molecular docking studies, where I can make a good expertise. These two parts lack a lot of important experimental details. So, at this stage the manuscript requires to be reconsidered after major revision.
- The 1st paragraph of the introduction should be illustrated with a Scheme of known approaches to benzoxanthones and a reported in this paper one.
- The reported compounds 3a-c are new ones, so the paper must contain in SI the pictures of their spectra (1H and 13 С NMR, DEPT135, IR).
- Description for molecular docking experiments should be given (program, method, what DNA was used and etc.).
Author Response
The manuscript “Synthesis, characterization and anticancer efficacy evaluation 1 of benzoxanthone compounds toward gastric cancer SGC-7901” describes investigations on anticancer action of a series of novel benzoxantone derivatives. Biological part of the paper is performed in a standard for the authors protocol, and I am not experienced enough to judge it. However, the manuscript contains chemical and molecular docking studies, where I can make a good expertise. These two parts lack a lot of important experimental details. So, at this stage the manuscript requires to be reconsidered after major revision.
1. The 1stparagraph of the introduction should be illustrated with a Scheme of known approaches to benzoxanthones and a reported in this paper one.
We have provided Scheme 1 including the approaches to benzoxanthones in the introduction.
2. The reported compounds 3a-c are new ones, so the paper must contain in SI the pictures of their spectra (1H and 13 С NMR, DEPT135, IR).
The 1HNMR, 13C NMR and HRMS spectra of the compounds have been provided in the supplementary materials.
3. Description for molecular docking experiments should be given (program, method, what DNA was used and etc.).
The procedure for molecules docking experiments have been added.

Round 2
Reviewer 1 Report
After the introduced corrections and supplements, the manuscript is ready for publication.
Reviewer 4 Report
The authors have made all the necessary changes. Now the manuscript can be accepted.